# From Traditional Dairy Product “Katak” to Beneficial *Lactiplantibacillus plantarum* Strains

**DOI:** 10.3390/microorganisms11122847

**Published:** 2023-11-23

**Authors:** Lili Dobreva, Dayana Borisova, Tsvetelina Paunova-Krasteva, Petya D. Dimitrova, Venelin Hubenov, Nikoleta Atanasova, Ivan Ivanov, Svetla Danova

**Affiliations:** 1The Stephan Angeloff Institute of Microbiology, Bulgarian Academy of Sciences, 26 Acad. G. Bonchev Str., 1113 Sofia, Bulgaria; ldobreva@microbio.bas.bg (L.D.); daqanara@abv.bg (D.B.); pauny@abv.bg (T.P.-K.); pdimitrova998@gmail.com (P.D.D.); vhubenov@microbio.bas.bg (V.H.); nikoletaatanasova21@gmail.com (N.A.); 2National Center of Infectious and Parasitic Diseases, bvd. “Yanko Sakazov” 26, 1504 Sofia, Bulgaria; ivanoov@gmail.com

**Keywords:** fermented milk products, postbiotics, *Lactobacilli*, *Lactiplantibacillus plantarum*, antimicrobial activity

## Abstract

Traditional milk products, widely consumed in many countries for centuries, have been drawing renewed attention in recent years as sources of bacteria with possible bioprotective properties. One such product for which only limited information exists is the traditional Bulgarian “katak”. This fermented yogurt-like product, renowned for its taste and long-lasting properties, possesses specific sensory characteristics. In this study, 18 lactic acid bacteria (LABs) were isolated from artisanal samples made in the Northwest part of Bulgaria. A polyphasic taxonomic approach combining classical phenotypic and molecular taxonomic methods, such as multiplex PCR, 16S rDNA sequencing, and MALDI-TOF MS, was applied, leading to the identification of 13 strains. The dominance of *Lactiplantibacillus plantarum* was confirmed. In vitro tests with the identified strains in model systems showed a promising broad strain-specific spectrum of activity against food-borne and human pathogens (*Staphylococcus aureus*, *Pseudomonas aeruginosa*, and *Escherichia coli*). Non-purified *Lactobacillus* postbiotics, produced during fermentation in skimmed and soya milks and in MRS broth, were estimated as limiting agents of virulence factors. The LAB’s production of lactate, acetate, and butyrate is a promising probiotic feature. A further characterization of the active strains and analysis of the purified post-metabolites are needed and are still in progress.

## 1. Introduction

Yogurt and cheese have been a staple of the Bulgarian cuisine since prehistoric times. They attract scientific interest with their excellent organoleptic characteristics, combined with nutritional value and safety [1,2]. Recently, such an interest has grown as a function of the desire for bioprotective agents that can naturally preserve food without the need for artificial preservatives. Several species of lactic acid bacteria (LABs) have been characterized as promising food protection agents. Due to their abundance in the environment and presence in raw foods, LABs may act as initiators of natural fermentation [3,4]. In fermented products (in the relevant matrix, fruits, cereals, vegetables, and dairy), a high number of LABs are found, and depending on the type of product and other factors (e.g., the dynamics of fermentation), they can vary from 4–5 log_10_ to over a billion living bacteria [5]. During fermentation, they produce a wide range of metabolites with a broad spectrum of activity. Postbiotics (or metabiotics/postmetabolites), such as produced active metabolites, low-molecular-mass ingredients, derived from food matrices and/or cell-derived fragments, enzymes, have attracted the attention of microbiologists in the last decade. The inhibition of bacterial growth by the action of substances derived from other microorganisms is a well-known barrier mechanism for various environmental niches and a generally accepted criterion in the selection of potential probiotic strains [6,7].

In addition, the demand among consumers for functional foods and health-promoting factors, such as probiotics (pro—for; bio—life) has increased. “Katak”, like other traditional fermented foods, is rich in autochthonous lactic acid microbiota [8]. However, there is scarce information on the taxonomic diversity and biological activity of the LAB microbiota of “katak”. Initial data shows the presence of *lactobacilli* [9,10]. The genus *Lactobacillus* (former name), proposed by Beijerinck in 1901 [11], which belongs to the *Lactobacillaceae* family, is the largest in the LAB group, with 533 species counted [12]. At present, *Lactobacillus* representatives are undergoing major taxonomic changes, according to which the genus has been divided into 25 genera, with some of the species gaining the status of separate genera [13]. From a scientific point of view, the biodiversity of LAB microbiota of such artisanal products is a challenge with practical significance. Previous studies have demonstrated the high viability and dominance of *lactobacilli* with promising antimicrobial potential from other rural regions [9].

In the present work, we study artisanal samples from a specific rural area—near the town of Lukovit. They were created without the addition of starters/live bacterial additives. This recipe of “katak” differs from that in the Southern part of Bulgaria and thus requires a separate investigation.

In order to determine the autochthonous lactic acid microbiota of this specific recipe of “katak”, the newly isolated *lactobacilli* are identified at the species level. The aim is to determine the species contributing to the safety and quality of this fermented milk product. In order to characterize them as natural antagonists of pathogens, with a role in “katak’s” long shelf life, in vitro testing is performed. Our work is part of a long-term targeted program to isolate and study new LAB strains as likely beneficial bacteria for health and food safety purposes.

## 2. Materials and Methods

### 2.1. Home-Made Samples of “Katak” and LAB Microbiota Characterization

Four samples of “katak” were obtained in 2021 from a small farm in the Balkan Mountain near Lukovit city [14]. They were produced from whole sheep’s milk at the end of the lactation period, according to an authentic recipe, without any starters and only salt added. The samples were stored at 10 °C for 2 months. A LAB isolation and microbiological analysis (CFU/mL) were conducted using MRS agar (Merck, Darmstadt, Germany) and Rogosa (HiMedia, Mumbai, India). Following the classical microbiological method (Koch method), 1 g from each collected sample was used for decimal dilutions. The isolated, pure cultures from the single colonies were obtained on agar medium (Rogosa agar) after 48 h at 37 °C. They were re-cultured twice in MRS broth (Merck, Darmstadt, Germany) with a pH pf 6.5, in anaerobic conditions (BD BBL™ GasPak, BD, Franklin Lakes, NJ, USA). Initial characterization by classical Gram stain, catalase, and oxidase tests (HiMedia kit) allowed the preliminary selection of LABs. Pure LAB cultures were stored at −20 °C in MRS broth supplemented with 20% (*v*/*v*) sterile glycerol (Merck).

### 2.2. Molecular Identification of Newly Isolated Lactic Acid Bacteria

DNA isolation: the microbial DNA extraction of isolates was performed with a HiPurA™ Genomic DNA Purification Kit (HiMedia, Mumbai, India), according to the manufacturer’s instructions. The obtained DNA was visualized in 1% *w*/*v* agarose gel (Sigma-type II, Sigma-Aldrich, Saint Louis, MO, USA) and applied as a template for PCR analyses.

PCR analysis and sequencing: Illistra^TM^ PuRe Taq^TM^ Ready To Go^TM^ PCR beads (Amersham Biosciences, Amersham, UK) were used in PCR protocol for the amplification of 16S rRNA gene and sequencing. The PCR products, obtained with universal primers (27F and 1492R, purchased from Macrogen Europe, Amsterdam, The Netherlands) were subjected to a standard sequencing procedure at Macrogen (Europe). The obtained sequences were manually processed on Applied Biosystems Sequence Scanner Software v1.0 to exclude primer-binding sites. The processed sequences were checked for similarities to the existing nucleotide database in the NCBI using BLAST software (online version) and deposited in NCBI GenBank.

Multiplex-PCR amplification with primers targeting the rec-A gene, according to Torriani et al. (2001) [15], was performed on a PCR cycler (Techne, UK), as previously described [16]. The DNA samples isolated from the type strains were used as positive PCR controls for multiplex PCR analyses: *Lactiplantibacillus paraplantarum* ATCC 700211^T^ *Lactiplantibacillus plantarum* ATCC 14917^T^, and *Lactiplantibacillus pentosus* ATCC 8041^T^.

Matrix-assisted laser desorption ionization-time flight mass spectrometry analyses (MALDI-TOF MS) were performed for 13 isolates. Briefly, a single colony from an exponential pure culture was deposited on a polished-steel MSP 96 target and overlaid with 1 µL 70% formic acid, followed by 1 µL of a saturated-cyano-4-hydroxycinnamic acid (HCCA) matrix solution (Bruker Daltonics, Billerica, MA, USA). Mass spectra were collected using Microflex LT, analyzed with Maldi Biotyper and the MBT Compass reference library (version 2022), which contained 4274 species (Bruker Daltonics).

### 2.3. Antibacterial and Antibiofilm Activities

A modification of the agar diffusion method was used, following Tagg and Mac Given, 1971 [17], in different variants: (i) tested cell-free supernatants with an acidic pH (marked as CFS/aCFS), (ii) neutralized cell-free supernatants (nCFSs) from LAB strains, cultured in MRS broth (24–48 h, 37 °C), and (iii) fermented milks (skimmed and soya). Both variants of LAB CFSs were collected after 24 or 48 h fermentation periods in MRS broth (HiMedia, pH 6.5) in a NUVE EN 400 (Ankara, Turkey)incubator at 37 °C and were used for the initial screening of antimicrobial activity. The LAB cells were harvested by centrifugation (5000 rpm, Hermle, Germany, for 10 min). The supernatants were obtained by centrifugation in the same conditions and filtered through a 0.22 µm bacterial cellulose acetate filter (MS^®^ CA Syringe Filter, Membrane Solutions). The CFSs were neutralized to pH = 6.0 ± 0.2 with 5M NaOH. Freshly prepared sterile soya milk and 10% (*w*/*v*) rehydrated skimmed milk (Scharlau, Spain) were sterilized (at 110 °C 15 min) and inoculated with 2% *v*/*v* of *Lactobacillus* culture for 24 h (MRS, 37 °C). As negative controls, sterile, non-fermented milks (skimmed and soya) and the corresponding media MRS or Rogosa (without inoculated LABs) were used for the assessment of the activity of CFSs and whey fractions from the milks. The activity of the tested CFSs was compaFred with the positive controls (antibiotics).

The test microorganisms used (Table 1) were stored at −20 °C in cultures supplemented with glycerol (20% *v*/*v*), or a lyophilized form, according to the provider (Aquachim LTD, Sofia, Bulgaria). Prior to the tests, they were pre-cultured twice in the corresponding broth media (Table 1) at 37 °C for 24–48 h. Their exponential overnight cultures, standardized according to 0.5 × McFarland standards, were used to inoculate (1% *v*/*v*) tempered agar in Petri dishes with the medium for the respective pathogen to obtain a final concentration of 10^6^–10^7^ CFU/mL. Aliquots (50 µL) of CFS/LAB cultures/fermented skimmed milk (10% *w*/*v*), reconstituted milk, or soy milk were added to wells in the agar medium with a diameter of 6 mm. After incubation for 24–48 h, the antimicrobial activity was reported by measuring non-growth zones (mm) around the well in triplicate.

To evaluate the effectiveness of the CFS (10% *v*/*v*) against pathogens, growth and biofilm inhibition were monitored spectrophotometrically. An optimized in vitro protocol, using 96-well microplates, was designed and overnight cultures of the pathogen, −0.5 × McFarland standard diluted 100× to obtain ~10^7^ CFU/mL, were used as the inoculum.

The procedure Crystal violet test detailed by Soto et al. (2006) [19] was employed for monitoring the pathogenic biofilm development in the 96-well microplate (6 wells/sample). The biofilm was formed during the pathogen’s growth in M63 medium comprising the following components: 0.02M KH_2_PO_4_, 0.04 M K_2_HPO_4_, 0.02 M (NH_4_)_2_SO_4_, 0.1 mM MgSO_4_, and 0.04 M glucose. An overnight bacterial culture was diluted in an M63 medium at a 1:100 ratio, serving as the initial inoculum for the biofilm experiments. The CFSs being tested were mixed with the inoculum in a tube and then applied to 96-well, U-bottomed, polystyrene microtiter plates (Corning, Corning, NY, USA) with a final volume of 150 µL. Additionally, the control sample included the bacterial inoculum in M63 with the addition of 10% (*v*/*v*) MRS broth. The plates were incubated under stationary conditions for 24 h at 37 °C. Following the incubation, the wells were washed 3 times with phosphate-buffered saline (PBS, pH 7.2), and the attached bacteria were stained using a 0.1% aqueous solution of crystal violet for the duration of 15 min. Subsequently, the wells underwent further washing with PBS, and the stained biofilm was dissolved using 70% ethanol. The optical density was measured at a wavelength of 570 nm using an Elisa Plate Reader (INNO, Seongnam-si, Korea).

### 2.4. Test for the Evaluation of the Swimming Motility of Pseudomonas aeruginosa PAO1

To evaluate the effects of LAB CFSs on the motility of *P. aeruginosa* PAO1, tryptic soy broth (Oxoid, UK) with 0.3% (*w*/*v*) Bacto agar (Difco) was used for the evaluation of motility-type swimming. The CFSs, 5–10% (*v*/*v*), of exponential *Lactobacillus* cultures were added to each Petri dish. Pure MRS medium (Hi Media, Mumbai, India) added (5–10% *v*/*v*) to the Petri dishes with TSB agar was used as the control. An overnight culture of *P. aeruginosa* PAO1 was placed at the center of the Petri dishes (seeded per spike). The plates were cultured at 37 °C and changes in the diameters of the motile zone of each colony were recorded periodically. The experiments were performed in triplicate and compared with the motility of the *P. aeruginosa* PAO1 in the control Petri dishes. The percentage of inhibition for the motility zone of *P. aeruginosa* was calculated according to the following formula:(Diameter of the colony in the control Petri − Diameter of the corresponding sample/Diameter of control sample) × 100

### 2.5. In Vitro Evaluation of Anti-QS Activity of CFS from LAB by Agar Diffusion Assay

An in vitro assay by the agar-well diffusion method on Petri dishes with tryptic soy agar (TSA) was carried out. The test culture *Chromobacterium violaceum* strain 30191 (DSMZ) was obtained in tryptic soy broth (TSB) at 30 °C for 24 h. Prior to the assay, the test culture was densitometrically (Densilameter II) calibrated to the McFarland standard to 1 × 10^9^ CFU/mL in a sterile physiological solution, and 1 × 10^5^ CFU/mL was inoculated in TSA Petri dishes. In each well, 50 μL of the sterile, spent culture supernatants (CFS and neutralized CFS) were dropped. In order to diffuse the substances in the agar, the Petri dishes were incubated at 4 ℃ for 2 h, followed by a 30 °C incubation for 24 h. The appearance of clear zones around the wall was monitored and measured in triplicate.

### 2.6. Gas-Chromatographic Analyses of Produced Volatile Fatty Acids (VFAs) in Growth Media

The concentration of volatile fatty acids (VFAs) was determined in the spent cultural media samples. One ml of the sample, i.e., the spent cultures (24 h of fermentation in MRS broth or HiVeg MRS broth, India), was acidified with about 50 µL 37% H_3_PO_4_ (until the sample pH reached 2.0). After 1 hour, the sample was centrifuged at 15,000 rpm (Hermle Z356K, Germany) for 10 min and an aliquot of the supernatant was mixed with an equal volume of 1% 2,2-dimethyl-butyric acid (as the internal standard). A total of 1 microliter of the final mixture was manually injected into a Thermo Scientific gas chromatograph (Focus GC model) equipped with a Split/Splitless injector, TG-WAXMS A column (length of 30 m, diameter of 0.25 mm, film thickness of 0.25 μm), and a flame ionization detector (FID).

### 2.7. Statistical Analysis

The values obtained from all the experiments (in triplicate for the agar well diffusion method, for swimming activity inhibition assessment, and in six replicates for Pseudomonas growth and biofilms inhibition) were presented as mean ± standard error bars (SE). Information on the antimicrobial activity was collected for each sample in triplicate and shown as the mean ± standard deviation (S.D.). The data were subjected to a statistical analysis and the differences for the samples and controls were assessed using an ANOVA test: * *p* < 0.05; ** *p* < 0.01, and *** *p* < 0.001.

## 3. Results and Discussion

### 3.1. Characterization of the Autochthonous Microbiota of “Katak”

In the present work, the autochthonous LAB microbiota of the traditional Bulgarian milk product “katak” were studied. All collected samples of this curd/yogurt-like product, with a salted milky-acid taste, were homemade from ewes’ milk. They have a long shelf life (up to 12 months). At the time of the analysis, after 7 months of cold storage, they showed a notable vitality value of 0.7–1.0 × 10^6^ CFU/g and pleasant sensory qualities. From the countable number of colonies on Rogosa SL agar plates (HiMedia, Thane, India), 18 pure cultures were isolated. They were identified as *lactobacilli*, using classical taxonomic methods. The morpho-physiological characteristics, Gram (+) staining, catalase- and oxidase-negative activity, and rod-shaped cell morphology of the strains were determined for the initial identification.

Then, 13 out of the 18 newly isolated strains (noted L1–L14) were identified at the species level using molecular genetic methods. The total DNA obtained from the exponential *Lactobacillus* cultures was used as a target for the molecular analyses: (i) 16S rDNA PCR analysis using universal primers (forward 27F and reverse 1492R), and (ii) sequence analyses and/or species-specific multiplex PCRs, MALDI-TOF. Four of the newly isolated strains were identified by the gold standard in bacterial taxonomy (16S rDNA gene sequencing), in Macrogen, Europe, Ltd., with a BLAST analysis, using the NCBI GenBank database; a high percentage of similarity was obtained for three closely related species: *Lactiplantibacillus plantarum*, *Lactiplantibacillus pentosus*, and *Lactiplantibacillus paraplantarum* (Table 2).

The data from the obtained 16S rDNA sequences for the four strains from “katak” (Table 2) place them in the *Lactiplantibacillus* genus *plantarum* group, despite the phenotypic and genotypic differences between them. Thus, an additional evaluation with an external automated Maldi-TOF MS system was performed (Table 2). The isolates L8, L9, L10, and L13 were successfully identified with a score > 1.9. They belong to the *Lactiplantibacillus plantarum* (formerly known as *Lactobacillus plantarum*) species with a high probability score.

A significant change in the taxonomy of the genus *Lactobacillus*, as well as in the phylogenetic group *L. plantarum* was made [13]. At present, *Lactobacillus plantarum, Lactobacillus pentosus*, and *Lactobacillus paraplantarum*, which share a similarity at 99.6% in the 16S rDNA gene, are grouped into a separate genus named *Lactiplantibacillus* with 14 other species [13].

MALDI-TOF MS was used as a rapid and reliable technique, which is commonly used for bacterial identification in clinical microbiology and research settings. An updated database with a wide range of bacterial species allowed us to confirm the correct identification of isolates from homemade “katak”. This accurate and reliable method has been used for LAB characterization purposes in different habitats, including French cheese [20] and oral microbiome [21]. The multiplex PCR with primers targeting the *rec*A gene generated specific PCR products of 318 bp corresponding to the *Lactiplantibacillus plantarum* species for 9 (L1, L2, L4, L5, L6, L7, L11, L12, and L14) out of 13 tested strains (Figure 1).

Multiplex PCR analysis is suitable for the differentiation of closely related species *L. plantarum*, *L. paraplantarum*, and *L. pentosus* and has been previously used to identify cheese isolates [16]. The same approach was applied for *L. plantarum* strains isolated from grape [22]. Our study added new information regarding the dominance of facultative heterofermentative LAB species in this product in correlation with the traditional recipe used and the rural region of production. As previously noted, this kind of “katak” is quite different from the homonymic product previously studied in another region of Bulgaria. It is also called “ahchak”, prepared only with an NaCl addition to ewe’s milk (no starters), and differs from the product produced in the Southern part of Bulgaria [1]. Therefore, obligate heterofermentative species reported by Tropcheva et al. (2014) were not isolated [9]. In the previous study of “katak” samples, the presence of *Lacticaseibacillus paracasei* and *Levilactobacillus brevis* was revealed [9]. Isolates with broad-spectrum antimicrobial properties from such a “katak” sample were identified as *L. brevis* and a predominance of heterofermentative *lactobacilli*, such as *L. brevis*, *Limosilactobacillus fermentum*, *Lacticaseibacillus casei*, and *Lacticaseibacillus rhamnosus*, was observed [9].

According to our data, for the first time, the stable dominance and presence of *L. plantarum* species were demonstrated for the variety of “katak” we studied [23]. Different recipes and specific geographic regions are likely to be responsible [23,24]. Moreover, *L. plantarum* is a highly adaptable, widespread, and autochthonous species in fermented food, and is able to colonize various ecological niches, including vegetables, meat, and gastrointestinal tract. It is an indigenous species in raw milk and non-starter microflora are involved in the cheese-ripening stage [25]. With its rich genome, the species probably contributes in a specific way to the sensory characteristics and long shelf-life of “katak”.

The task of identification was important from a scientific perspective and also in view of the European Food Safety Agency’s regulations. The newly isolated strains may also prove useful as probiotics.

### 3.2. Newly Isolated Lactobacilli from “Katak” as Active Antagonists of Pathogens

Combating pathogens has always been the focus of microbiology. Due to the risk of rapidly increasing antibiotic resistance, new approaches to constrain it are constantly being researched. With this aim, the in vitro activity against different pathogens was assessed.

#### 3.2.1. Anti-Microbial Activity of *L. plantarum* Strains from “Katak” against *E. coli*

An initial screening of the anti-microbial activity of the identified *L. plantarum* strains (against *E. coli* strain 420 and strain TU5) was performed. The postmetabolites produced during the fermentation in MRS broth of soya and skimmed milks were assessed. A strain-specific spectrum of activity was shown by the agar well diffusion method (Figure 2). The first step to characterize the nature of active postmetabolites was a comparative analysis of the acid and neutralized (with 5N NaOH) spent cultures. Strain-specific activity against *E. coli* 420 for *lactobacilli* from “katak” cultured in MRS broth, sterile reconstituted 10% *w*/*v* skimmed milk (with the exception of L1 and L7), and soya milk was determined. This activity ranged from moderate to high. The inhibitory compounds produced during fermentation in the skimmed and soya milks were also promising (Figure 2A). No activity was detected from the MRS broth and sterile milk used as the controls. Therefore, the significance was calculated against a positive control—the zone generated by Ciprofloxacin (5 µg/disk). Even though the inhibitory zones for soya and 10% *w*/*v* skimmed milks were smaller than those for CFSs, the results show that the tested *lactobacilli* have the potential to be implemented as bioprotective additives for dairy and non-dairy fermented foods.

In the last decade, a large group of active LAB strains from Bulgarian fermented milk products were characterized [1]. The *L. plantarum* strain L2 showed greater activity than the used positive-control lactic acid (Merck)—3.3% against a clinical multi-resistant strain isolated from a patient with recurrent UTI infections (Figure 2B).

For every analysis, the mean and standard deviation (mean ± S.D.) was calculated, and the SD values were displayed as Y-error bars. Statistical significance (** *p* < 0.01; and *** *p* < 0.001, sample numbers n = 3) of all the tested samples (A) v/s Erythromycin (as the positive control) and NS in the variant (B) v/s lactic acid (3.3% *v*/*v* control) were displayed with the ANOVA test.

Active LAB metabolites may prevent the spoilage and overgrowth of pathogenic microflora. Notably, *E. coli* 420 was highly sensitive to tested postbiotics (48 h CFS) obtained during the fermentation of *L. plantarum* strains, as demonstrated for 12 out of the 13 strains studied. The measured zones ranged between 40–85% of those of the chloramphenicol antibiotics (30 µg/disc) used as the control (Figure 2A). The number of active strains (*L. plantarum* L1, L5, L6, L7, and L8) in the test with neutralized CFSs against this pathogen was much lower (Figure 2A). Two of the active strains, L2 and L4, as living cells and CFSs (Figure 2B) inhibited *E. coli* TU5—an outpatient clinical strain resistant to four of the widely used antibiotics. Thus, a possible synthesis of bacteriocin-like inhibitory substances (BLISs) or other metabolites, different from acids, were produced. Furthermore, the similar beneficial effects of postbiotics may be beneficial to healthy gut homeostasis in vivo.

#### 3.2.2. Initial Characterization of Metabolites Produced during the Fermentation in MRS Broth

In naturally fermented dairy products, *Lactobacillus* spp. are able to inhibit and control pathogen proliferation [26]. Due to their metabolism, they are able to produce different metabolites that can modify the environment and exhibit inhibitory effects on other microorganisms [27]. These include hydrogen peroxide, organic acids (lactic, acetic, etc.), bacteriocins, bacteriocin-like substances, reuterin, diacetyl, and others [28]. The initial gas chromatographic analysis confirmed specific differences in the produced metabolites (Table 3) for our *L. plantarum* isolates from “katak”.

*L. plantarum* L5 showed a high production of acetate, which is one of the important metabolites for healthy gut homeostasis (Table 3). Such VFAs (acetic, propionic, and butyric), produced through fermentation in vivo, may lower the pH in the gut lumen, making it more acidic. Acetic, propionic, and butyric acids may cause microbial inhibition. Thus, they may inhibit the growth and survival of many pathogenic bacteria that prefer a more neutral or alkaline pH. The same beneficial mechanism used by LABs prevented the development of spoilage bacteria [29,30], potentially pathogenic species [31], in traditional dairy products.

The activity was due to the lactic and other VFAs produced, and H_2_O_2_. According to Lanciotti et al. (2003), *L. plantarum* strains produce diacetyl, which is known to inhibit *E. coli*, *S. aureus*, and *Listeria monocytogenes* [32]. A positive correlation between lactic acid concentration and antimicrobial activity was found for *L. rhamnosus* GG against the *S.* Typhimurium strain. As a facultative heterofermentative bacterium, *L. plantarum* produces lactic acid and ethanol or acetic acid (Table 2), which lowers the pH of the medium. The low pH inhibits the sensitive microorganisms and makes the organic acids liposoluble so that they reach the cytoplasm by crossing the cell membrane [33]. Acetic and propionic acids cause protein denaturation and intracellular acidification [34].

#### 3.2.3. Antagonistic Activity of *L. plantarum* Strains against *Staphylococcus aureus*

The in vitro experiments against *Staphylococcus aureus* V00037 proved that, within the species *L. plantarum*, there are strains with different spectra of inhibitory activity. Due to its metabolic activity, the results are the most pronounced for the CFS produced in the MRS broth. *Lactobacilli* showed approximately the same zones of inhibition in the initial test of *lactobacilli* cultures in the MRS medium by the agar diffusion method, except for *L. plantarum* L13 (Figure 3). However, when cultured in sterile, freshly prepared soya milk, only three strains showed activity (Figure 3A). It is evident that in all three versions of the test (with CFS, skimmed milk, and soya milk), the obtained zones of inhibition are much smaller than the positive-control-antibiotic Erythromycin (15 µg/disk, NCZPD, BulBio). Despite the considerable statistically significant difference (*p* < 0.05) between them, these results confirm the synthesis of postbiotics with promising activity.

We further assessed their inhibitory effect by following the pathogenic growth inhibition with a spectrophotometer INNO (at 600 nm), using a model microplate system for up to 24 h. The growth of *S. aureus* in the presence of CFSs (from the exponential 24 h and stationary phases—48 h of 13 tested LAB cultures) was differently affected, shifting from a strong inhibition to stimulation, as was the case for strain L11 (Figure 3B). Strains L9 and L10 presented inhibitions in both variants.

*Lactobacilli* can exert strain-specific metabolic activity, and greater activity was shown for CFSs obtained from the stationary phase (48 h). The effect was probably due to acidity, which was higher in this phase (pH 3.9–4.5). The activity of living bacterial cultures and cell-free supernatants of *L. plantarum*, isolated from Polish regional cheeses, inhibited the growth of *S. aureus* [35]. Setyawardani et al. (2014) [36] determined higher activity for Gram (−) compared to Gram (+) bacteria. This is consistent with our results, where the activity towards *S. aureus* V00037 (average: 4–5 mm zones) was weaker compared to the activity towards *E. coli* 420 (average: 8–17 mm) and *E. coli* TU 5 (average: 4–6 mm inhibition zones) (Figure 2A). Therefore, “katak”, like other fermented dairy products, may be a promising source of new active strains. Goat milk-based products have been reported as a source for the isolation of active *L. plantarum* against *E. coli*, *P. aeruginosa*, *S. typhimurium* [37,38], and *S. aureus* [38], as have those from raw sheep’s milk [39].

The assembled data (Figure 2 and Figure 3) clearly show that *L. plantarum* from the “katak” are characterized by a strain-specific broad spectrum, with the largest group in terms of the number of active strains against *E. coli*. In vitro or in situ metabolites produced during various fermentation processes of such LAB cultures can successfully protect against the invasion of pathogens and/or food spoilage microbes.

#### 3.2.4. Can Postbiotics Produced from *L. plantarum* Strains Inhibit Virulence Factors and Quorum-Sensing Controlled Pathogenesis?

In vitro effects of LABs on the growth, biofilms, and motility of *Pseudomonas aeruginosa* PAO1.

The pathogen invasion of host cells is mostly mediated by virulence factors. Promising results with antimicrobial activity against *E. coli* strains (see Section 3.2.1) allowed us to perform an in vitro assessment of other barrier mechanisms of LABs against pathogens. The *P. aeruginosa* strain PAO1 from the International Reference Panel [18] was used as an extensively applied microorganism to understand biofilm formation and its associated virulence factors. The growth inhibition of *P. aeruginosa* (OD at 600 nm) was monitored in the presence of 10% *v*/*v* produced postbiotics at different time points (2, 4, 18, and 24 h), in different model media: (i) MRS-exponential cultures in MRS broth (pH 6.5); (ii) vMRS-HiVeg MRS (vegan variant of MRS, pH 6.5), and (iii) whey fractions fermented with 10% (*v*/*v*) skimmed milk (Sharlau, Barcelona, Spain). Growth of *P. aeruginosa* PAO1 in TSB supplemented with sterile MRS broth was used as a control.

All tested postbiotics from LAB cultures, with 10% (*v*/*v*) added to the TSB, significantly inhibited the growth of *P. aeruginosa* at 2, 4, 18, and 24 h cultivation periods in TSB broth. The CFSs from vegan MRS for *L. plantarum* L1, L8, L11, and L12 possessed greater activity, while the strains L10 and L14 were producers of active postmetabolites during the fermentation in MRS broth. Probably due to the high lactic acid production during the milk fermentation process, all tested whey fractions showed a significant inhibition (more than 40%), which was as high as the inhibition of the CFSs from the MRS and the vegan variant of the MRS broth (Figure 4).

The heatmap diagram reflects the impacts of different postbiotics tested against pathogenic bacteria (Figure 4). The color key indicates the growth inhibition of *P. aeruginosa* PAO1 during cultivation (at 2, 4, 18, and 24 h) in tryptic soy broth, supplemented with 10% *v*/*v* acid CFS produced from exponential LAB cultures (in MRS broth, vegan MRS, and WF). Percent of inhibition was calculated vs. control (TSB supplemented with corresponding LAB media).

A peculiar form of counteraction and containment of the pathogen was observed in the parallel assays for the biofilm formation of *P. aeruginosa* PAO1 in the TSB in the presence of 10% (*v*/*v*) CFS at 24 h. The crystal violet (CV) test was used to assess the biofilms formed during the 24 h cultivation of *P. aeruginosa* PAO1 in a 96-well microplate in the TSB using a control—the biofilm formed in TSB + 10% (*v*/*v*) MRS broth. Promising in vitro anti-biofilms effects were observed with the assessment (in triplicate) of the tested LAB postmetabolites from *L. plantarum* strains (Figure 5).

The heatmap shows the impact of different postbiotics on *Pseudomonas* biofilms (Figure 5). The color key indicates the biofilms’ inhibition of *P. aeruginosa* PAO1 during cultivation (at 2, 4, 18, and 24 h) in TSB supplemented with 10% *v*/*v* CFS * from exponential LAB cultures (in different media) calculated vs. control (TSB supplemented with corresponding LAB media).

The results of the in vitro CV test with biofilms (Figure 5) clearly show the high inhibitory potential of postbiotics produced during the fermentation of the tested *lactobacilli* in different culture media: (i) MRS exponential cultures in MRS broth (pH 6.5); (ii) vMRS-HiVeg MRS (vegan variant of MRS, pH 6.5), and (iii) whey fractions fermented with 10% (*v*/*v*) skimmed milk (Sharlau, Spain). All tested postbiotics from the LAB cultures, added at 10% *v*/*v* to the TSB broth, significantly inhibited the *P. aeruginosa* biofilms. The anti-biofilm effects were strain specific, possibly due to the considerable lactic acid production during fermentation; some of the tested whey fractions exhibited greater anti-biofilm forming activity in comparison with the CFS from the MRS and the vegan variant of MRS broth. Shokri D. et al. (2018) [40] demonstrated two *Limosiactobacillus fermentum* among the 57 isolates tested, with broad inhibition/killing and anti-biofilm effects against pan-drug-resistant (PDR), extended antibiotic-resistant (XDR), and multidrug-resistant (MDR) *P. aeruginosa* isolates. Their mode of action occurred due to the production of three VFAs—lactic, acetic, and formic. This was proven in vivo with the intranasal application of different heterofermentative species of *lactobacilli* able to prevent *P. aeruginosa* acute pneumonia in mice [41].

*Pseudomonas*’ biofilms are an important aspect of pathogenicity, as they provide protection and enhance resistance against antibiotics and the host’s immune response. Biofilm formation is particularly relevant in the context of infections, as *P. aeruginosa* can cause various infections in humans.

The results obtained from the metabolomics analyses of spent LAB cultures (Table 2) confirm the presence of metabolites with antibacterial and anti-biofilms activities. A similar promising effect on biofilms was observed in vitro with the 13 *L. plantarum* strains co-cultured with *P. aeruginosa* PAO1. We hypothesize that some of the *lactobacilli* tested may produce bacteriocin-like substances (BLISs). The reason is the reported activity of the neutralized CFSs (Figure 2A). Additional characterizations, however, are needed. Synergic effects may probably be supposed, due to the observed additional activity of other factors of *Pseudomonas* virulence, such as the inhibition of swimming motility. The ability of *lactobacilli* from “katak” to influence different motilities of the *P. aeruginosa* strain PAO1 was evaluated in a model system with tryptic soy agar (0.3% *w*/*v*). The results summarized in Figure 6 show the significant inhibition of the flagellar activity of the *P aeruginosa* strain PAO1, in the presence of CFSs, only in the early lag phase (4 h), while neutralized CFSs partially retarded the motility of the pathogen (Figure 6A,B).

The inhibition of more than 50% within 4 h decreased to only 8–10% relative to the control for some of the postbiotics tested (ex. produced by *L. plantarum* L1, L9, and L14). The strain-specific effects, however, have to be pointed out as more distinctly observed only in the early stages of *Pseudomonas* colony growth (Figure 6A).

The motility of some bacteria plays an important role as a determinant of the virulence of bacteria, which uses a swimming-type action to move to a specific surface and attach and then spread over the surface by swarming- and twitching-type movements. The latter movements are performed by multiple flagella and type-IV pili, respectively [42].

Swimming motility is required for *P. aeruginosa* in order to explore and colonize new environments, discover nutrients, and avoid adverse conditions. This motility mechanism plays a crucial role in the bacteria’s pathogenicity, as it helps the bacteria to reach and establish infections in various host tissues and organs. Spreading on surfaces is important for the colonization and establishment of biofilm communities of bacteria that are particularly resistant to antimicrobials [43]. For this reason, targeting bacterial motility is of particular importance for limiting surface colonization. In this regard, there is a growing interest in compounds capable of fulfilling this role. The in vitro motility test evaluates the ability of certain substances to have an effect on bacterial motility.

Anti-quorum-sensing (QS) effects of CFSs from *lactobacilli* on violacein synthesis.

The search for inhibitors of the QS mechanism is aimed at limiting cells’ communication and reducing virulence properties, rather than inhibiting the growth of microorganisms [44]. *C. violaceum* is a frequently used bacterial strain in the search for new quorum-sensing inhibitors [45]. The bacteria form violet colonies on different laboratory media and the coloring is provided by the pigment known as violacein. The vio operon encodes the synthesis of the pigment and is controlled by QS. To evaluate the anti-quorum-sensing activity of CFSs from LABs against biosensor strain *C. violaceum* 30191 (DSMZ), the agar diffusion method was applied. The results of the experiments reveal that some of the tested CFSs (6 out of 11) have the potential to inhibit violacein synthesis (Figure 7A). The absence of the purple pigment around the wells is an indication of inhibition (Figure 7B). The most promising effect was observed for CFSs from *L. plantarum* L2 cultures, where the zone of inhibition was 18.5 mm. The lowest inhibitory potential was assessed for L12 (9.5 mm zone) and no effect was observed for L4. In a study by Diaz et al. (2019) [46], *Lacticaseibacillus casei* CRL 431 and *Lactobacillus acidophilus* CRL 730 inhibited the violacein production of *C. violaceum* in a concentration-dependent manner of the supernatant extracts used.

The search for inhibitors of the QS mechanism has been directed towards inhibiting cell communication and reducing virulence properties, rather than inhibiting the growth of microorganisms [44]. Multiple studies have established the inhibition of the violacein pigment by different strains of *lactobacilli*. Only a few investigations have determined the compounds produced by lactic acid bacteria, which affect the virulence and chemical communication of pathogens. Patel et al. (2022) [47] reported the ability of unpurified *L. plantarum* biosurfactant to inhibit violacein pigment production at concentrations below MICs of 0.5, 1.5, and 2.5 mg/mL.

Relatively little is known about the nature of the postbiotics produced by lactic acid bacteria that influence the virulence and cellular communication of pathogens. QS inhibitors with synthetic or natural origins are a sought-after alternative to antimicrobial agents that, unlike antibiotics, do not affect growth and do not exert selective pressure towards the development of resistance [45,48]. Thus, they reduce the virulence of pathogens [44]. In this way, the development of resistance is avoided [49] and the pathogens may be restricted. The results presented add to the scientific information on *L. plantarum* as a widespread species with a promising spectrum of antimicrobial activity [50]. Natural fermentations in different food matrices may be a promising source of various postbiotics that, through different barrier mechanisms, limit the spread of pathogens and contribute to food safety [51].

## 4. Conclusions

The fermented yogurt/curd-like product “katak”, produced according to a traditional recipe in the Northwestern part of Bulgaria, possesses specific non-started autochthonous microbiota, dominated by the species *L. plantarum*. This is one of the species of *Lactobacillacea* with the broadest spectrum of antagonistic activity. The 13 newly identified species were active against Gram (+) and Gram (−) bacteria. The strain-specific inhibitory effects on the growth and biofilms formed in vitro by *E. coli* 420, *S. aureus*, and *P. aeruginosa* PAO1 were estimated. Produced active metabolites (VFAs) may contribute to the safety and long shelf life of this product, produced VFAs, or other active metabolites. The postbiotics produced during milk fermentation (in skimmed and soya milks) revealed the potential of characterized strains from “katak” to be advanced as bioprotective additives. The highly promising potential of selected *L. plantarum* presented effects on QS-controlled factors of virulence of *P. aeruginosa* PAO1. The possible production of BLIS, in addition to being such a useful bioprotective property, will determine its high probiotic potential. The spectrum of postbiotics produced in the course of fermentation may beneficially influence in vivo gut homeostasis. However, the characterization of these beneficial postmetabolites is necessary, and in combination with the evaluation of other functional characteristics is still in progress. From a scientific point of view, new data on the active postbiotics produced by the autochthonous microbiota of such fermented products are important before this old technology is further implemented in the modern-world setting.

## Figures and Tables

**Figure 1 microorganisms-11-02847-f001:**
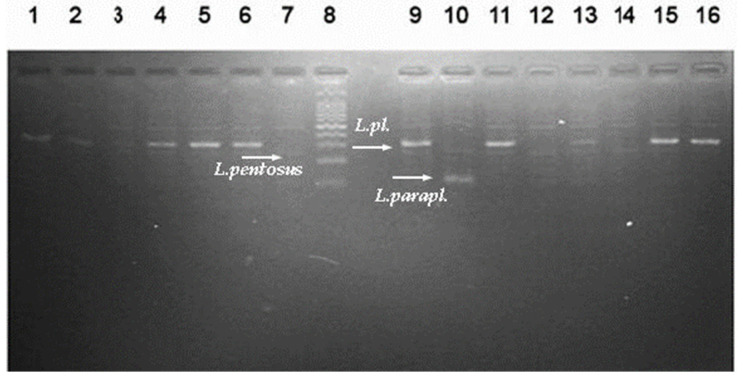
Multiplex PCR analysis of newly isolated LABs, with species-specific primers for *L. plantarum*, *L. paraplantarum,* and *L. pentosus*. The PCR products were visualized in 1.5% agarose gel (agarose sigma, type I) after ethidium bromide staining and the use of the transilluminator uVp system and camera Image Master VDS (Amersham Pharmacia Biotech, Italy). Lines: 1—L1; 2—L2; 3—L12; 4—L4; 5—L5; 6—L6; 7—*L. pentosus* ATCC 8041T; 8—Mol. -Gen Ladder 100 bp + 1.5 kbp (Gennaxon, Germany); 9—*L. plantarum* ATCC 14917^T^ (PCR control); 10—*L. paraplantarum* ATCC 700211^T^ (PCR control); 11—L7; 12—L8; 13—L9; 14—L10; 15—L11; 16—L13.

**Figure 2 microorganisms-11-02847-f002:**
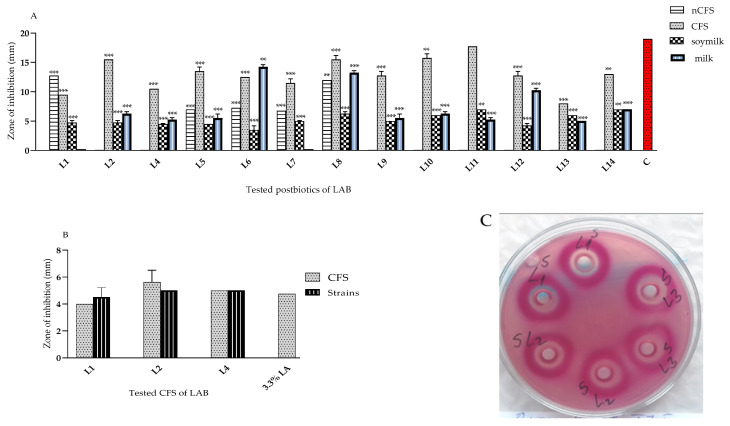
In vitro inhibitory activity of *L. plantarum’s* produced metabolites (CFS and neutralized CFS—nCFS) from exponential culture in MRS, fermented skimmed milk (10% *w*/*v* Sigma), and fresh-made sterile soya milk, and LAB cultures against: (**A**) *E. coli* 420, (**B**), and (**C**) *E. coli* TU5—a multi-resistant outpatient strain. Legend: all tests were repeated in triplicate by the agar well diffusion method using a McConkey agar plate. The diameter of the 6 mm well was removed. Two controls were used: antibiotic C (Ciprofloxacin) (BulBio, Sofia, Bulgaria)—5 µg/disc (**A**) and LA (lactic acid) (Merck)—3.3% (**B**).

**Figure 3 microorganisms-11-02847-f003:**
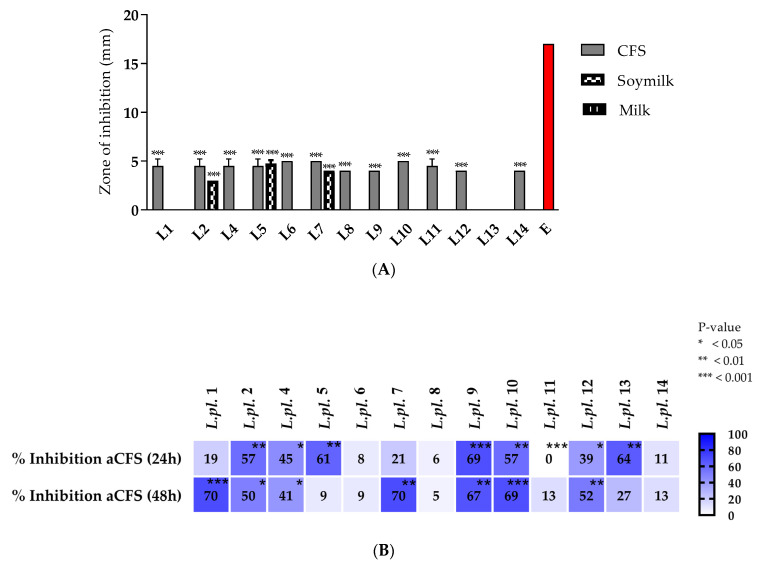
(**A**). In vitro inhibitory activity of *L. plantarum*’s CFS from exponential culture in MRS, fermented skimmed milk (10% *w*/*v* Humana, Germany), and fresh-made sterile soya milk against *S. aureus* V00037. Legend: for every analysis, the means and standard errors are calculated, and S.D. values are displayed as Y-error bars in the figures. Standard deviations (sample n = 3) are displayed as error bars. ANOVA test *p* < 0.05. A positive-control, E (Erythromycin), was used (15 µg/disk, NCZPD, BulBio). (**B**). In vitro inhibitory activity of *L. plantarum*’s postbiotics (aCFS) from exponential (24 h)- and stationary-phase (48 h) cultures in MRS broth against *S*. *aureus* V00037. The data from triplicates are subjected to statistical analyses and the differences for the samples and controls are assessed using the ANOVA test: * *p* < 0.05; ** *p* < 0.01, and *** *p* < 0.001.

**Figure 4 microorganisms-11-02847-f004:**
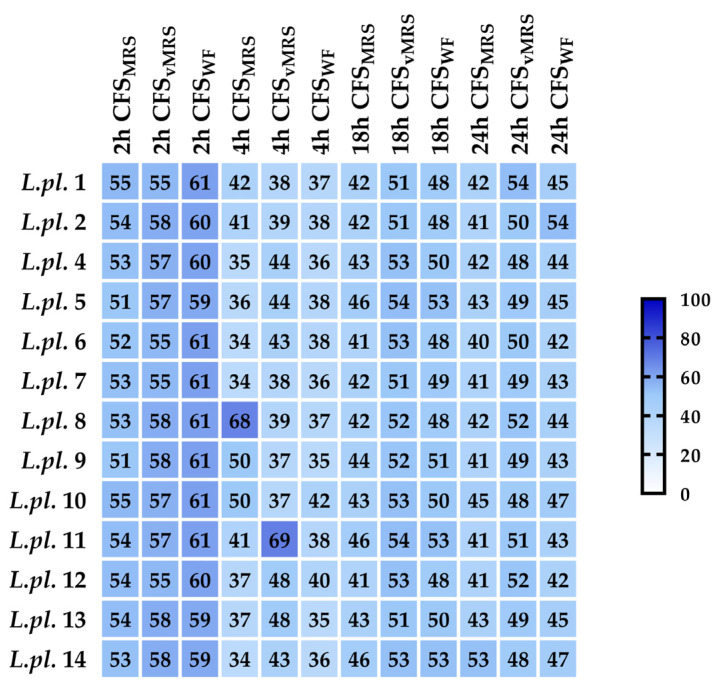
Heatmap diagram summarizing in vitro results of the inhibitory activity of Lactobacillus from “katak” on *P. aeruginosa* growth (calculated as % inhibition v/s control). The data from six replicates were subjected to statistical analyses and the differences for the samples (with postbiotics) and controls were assessed using the ANOVA test.All results show *p* < 0.001. Legend: horizontal: tested variants of postbiotics (CFSs) produced during the fermentation in different cultural media (MRS, vMRS, and WF from milk and time point in hours) of P. aeruginosa inhibition measurements. The sample codes on the right side consist of the abbreviation *L.pl*., derived from the name *Lactiplantibacillus plantarum* of the producers and strain numbers (L1 to L14).

**Figure 5 microorganisms-11-02847-f005:**
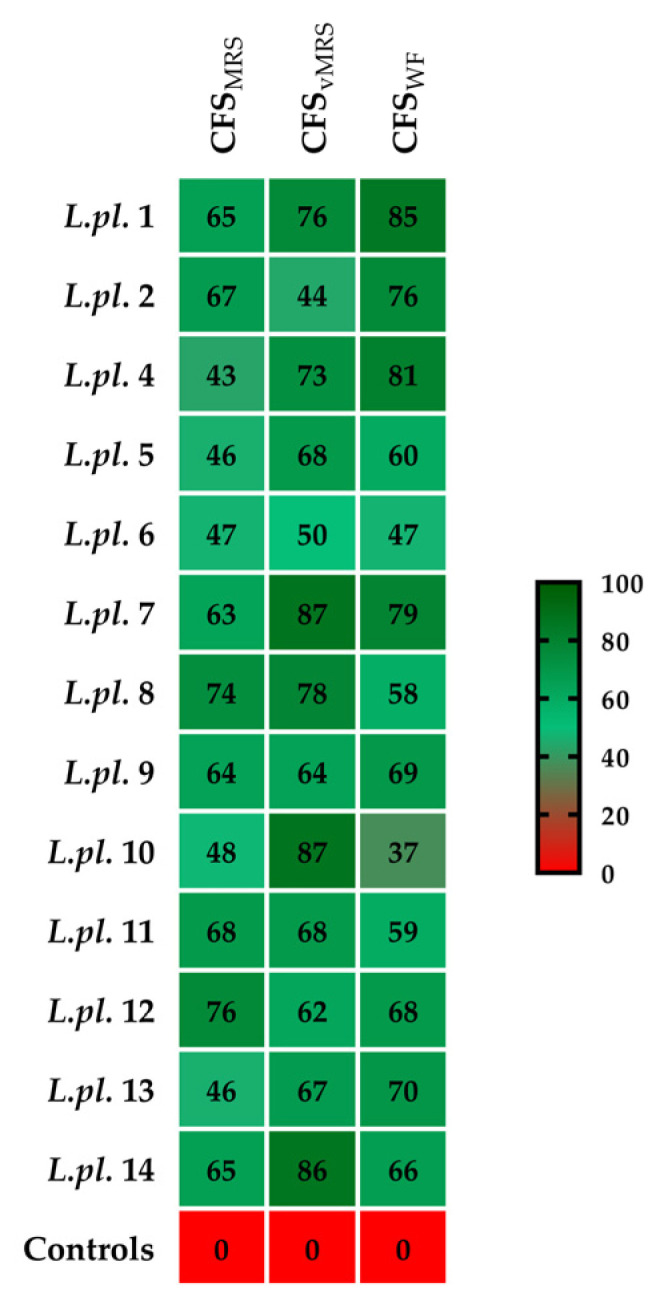
A heatmap diagram presenting the inhibition (as %) of *P. aeruginosa* biofilms formed in the presence of 10% (*v*/*v*) LAB postbiotics (CFSs from 24 h MRS broth, vMRS, and WF of fermented milks) tested. The data from six replicates were subjected to statistical analyses and the differences for the samples (with postbiotics) and controls were assessed using the ANOVA test: all results show *p* < 0.001. Legend: color code ranges from red (no inhibition as for the control) to green (higher % is dark green). The sample codes on the right side of the dendrogram consist of the abbreviation *L.pl.*, derived from the name of the producers *Lactiplantibacillus plantarum* and strain number (L1, L2, etc.).

**Figure 6 microorganisms-11-02847-f006:**
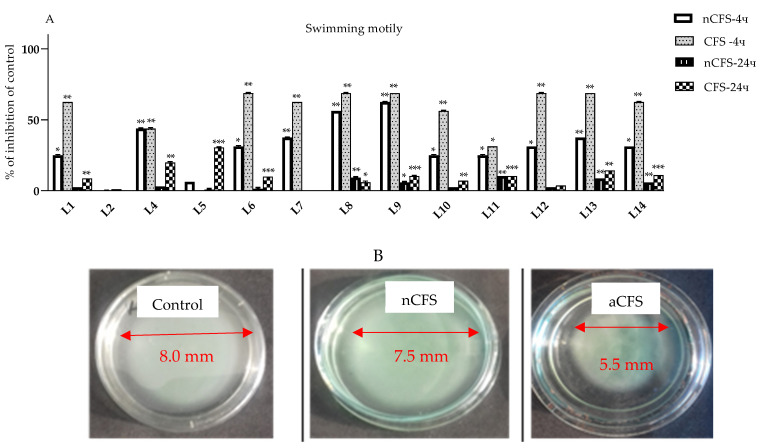
(**A**) In vitro test for the evaluation of flagellar activity (assessed for change in diameter during swimming motility in triplicate) of *P. aeruginosa* PAO1 in TSB with 0.3% agar (Oxoid, Hampshire, UK) in the presence of 1% (*v*/*v*) CFS and neutralized CFS (pH 6.0) of *L. plantarum* L1–L14 expressed as % inhibition relative to the control (no CFS added), and (**B**) illustration of the results from the in vitro test in the presence of postbiotics produced by *L. plantarum* strain L5 from “katak”. The data are presented as means ± SDs. * *p* < 0.05; ** *p* < 0.01, and *** *p* < 0.001.

**Figure 7 microorganisms-11-02847-f007:**
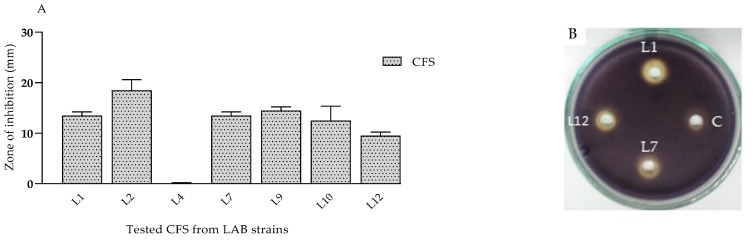
(**A**) In vitro determination of the ability of produced postbiotics (CFS (48 h cultures, MRS broth)) to inhibit QS controlled processes of violacein synthesis of *Chromobacterium violaceum* 30191 DSMZ, expressed as a clear zone (in mm), and (**B**) illustration of results from agar diffusion method used and results for 3 tested strains. The data are presented as means ± SDs. Legend: C—control MRS (Hi media).

**Table 1 microorganisms-11-02847-t001:** Test—microbes used for in vitro antimicrobial activity assessment.

No.	Test Microorganisms	Cultural Media
1	*Staphylococcus aureus* V00037	BHI broth (Difco) and BHI agar
2	*Escherichia coli* HB101 (IMicB collection)	BHI broth (Difco) and BHI agar, MacConkey (Sigma, Germany)
3	*Escherichia coli* clinical strain TU5	BHI broth (Difco) and BHI agar
4	*Escherichia coli* 420	BHI broth (Difco) and BHI agar
5	*Pseudomonas aeruginosa* PAO1 International Reference Panel (De Soyza et al. 2013) [18]	Tryptic soy broth/agar (HiMedia) M63 (for biofilm assays)
6	*Chromobacterium violaceum* strain DSMZ 30191	Tryptic soy broth/agar (HiMedia and Oxoid)

**Table 2 microorganisms-11-02847-t002:** Comparative molecular analyses for species identification of *lactobacilli* isolated from traditional milk product “katak”.

Strain	Molecular Identification of Isolates			MALDI-TOF MS Analysis	
	BLAST-16S rDNA Sequence—Organism (Best Match)	Similarity (%)	Species/NCBI Accession No	Organism (Best Match)	Score Value
L8	*L. pentosus* 124-2	99.79	*L. plantarum*/ OR528606	*L. plantarum*	2.14
	*L. plantarum* CIP 103151;	99.79			
L9	*L. plantarum* JCM 1149	100	*L. plantarum*/ OR528607	*L. plantarum*	2.00
	*L. pentosus* 124-2	100			
	*L. paraplantarum* DSM 10667	100			
L10	*L. plantarum* JCM 1149;	100	*L. plantarum*/ OR528608	*L. plantarum*	2.07
	*L. pentosus* 124-2	100			
L13	*L. plantarum* JCM 1149	100	*L. plantarum*/ OR528609	*L. plantarum*	2.16
	*L. pentosus* 124-2	100			
	*L. paraplantarum* DSM 10667	100			

**Table 3 microorganisms-11-02847-t003:** Summary of the results for the gas-chromatographic determination of VFAs of *L. plantarum* strains with antimicrobial activity.

CFS from LAB	VFA Component, g/L							Total VFAs, g/L
	Acetate	Propionate	i-Butyrate	Butyrate	i-Valerate	Valerate	Caproate	
*L. plantarum* L1	8.23	0.11	0.00	0.07	0.06	0.10	0.07	8.64
*L. plantarum* L2	8.65	0.13	0.00	0.07	0.06	0.18	0.06	9.15
*L. plantarum* L4	9.03	0.22	0.00	0.00	0.07	0.11	0.09	9.52
*L. plantarum* L5	16.97	0.59	0.00	0,10	0.34	0.22	0.18	18.40
*L. plantarum* L5 *	10.75	0.29	0.00	0.00	0.11	0.00	0.00	11.15
*L. plantarum* L6	8.70	0.24	0.00	0.00	0.07	0.10	0.00	9.11
*L. plantarum* L7	8.90	0.18	0.00	0.06	0.06	0.28	0.00	9.26
*L. plantarum* L8	9.50	0.11	0.00	0.06	0.06	0,00	0.06	9.79
C-MRS	7.07	0.10	0.00	0.09	0.08	0.09	0.00	7.44
C-Hiveg MRS *	4.45	0.10	0.00	0.00	0.08	0.00	0.15	4.78

Legend: C—controls: C-MRS. * Results from fermentation in HiVeg MRS (a vegan nutrient broth for *lactobacilli*).

## Data Availability

Data are contained within the article.

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
