# Peer review of "From Traditional Dairy Product “Katak” to Beneficial Lactiplantibacillus plantarum Strains"

_microorganisms, 2023, doi:10.3390/microorganisms11122847_

Round 1
Reviewer 1 Report
Comments and Suggestions for Authors
This study aimed to isolate lactic acid bacteria from the traditional dairy product “Katak”. The functions of the isolates were also evaluated by various analysis. This research is interesting and meaningful. The manuscript should be accepted after minor revision.
Line 40 lactic acid bacteria should be removed.
Line 51 what is difference between the lactic acid microbiota and LAB microbiota.
Line 55 Please check the new name of Lactobacillus.
Line 64 lactobacilli should be italic throughout the manuscript.
Line 132 the citation should be constant.
Author Response
Thank you for your kind comments, please see the attached reply.

Reviewer 2 Report
Comments and Suggestions for Authors
The manuscript is about the isolation of LAB from a traditional dairy product "Katak". I have some comments to the authors to consider for future experiments with LAB.
1. In order to call a LAB isolate a probiotic strain or possess postbiotics, there are a set of experiments that needs to be conducted. For example, acid and bile salts tolerance, adhesion assay, and antimicrobial agent characterization.
2. The scientific term "Postbiotics" is defined as "preparation of inanimate microorganisms and/or their components that confers a health benefit on the host". i.e. dead microorganisms or the remains of them after lysis.
Check this review: Ma L, Tu H, Chen T. Postbiotics in Human Health: A Narrative Review. Nutrients. 2023 Jan 6;15(2):291. doi: 10.3390/nu15020291. PMID: 36678162; PMCID: PMC9863882.
Therefore, it is better not to use this term to refer to CFS.
3. What is the number of isolated LAB strains? According to the abstract, 18 LAB isolates were isolated. However, in the last paragraph of the introduction, only 13 LAB isolates are mentioned. Why 5 isolates were ignored from subsequent experiments? In Table 2, only 4 isolates were identified. Why these four isolates only were identified?.
4. According to the materials and methods section, Katak is made by adding only salt to whole sheep's milk and incubation at 10℃ for 2 month. More details are needed. Based on the described conditions, I would expect milk putrefaction without milk boiling and addition of starter culture.
5. Line 90>> PCR binding sites??
6. Line 145>> Elisa Plate Rider?
7. Line 107>> Acidified and neutralized? There is no need to write "acidified" for untreated LAB cell free supernatant.
8. Line 121 & 347>> Standard growth curve is needed for each isolate to determine exponential/stationary stage. For example, I expect many LABs to be in stationary phase after 24 h.
9. Line 132/133>> "the biofilm was cultured….". How?
10. Materials and Methods section experiment 2.4>> Why the motility? I wonder if any effects were observed on the flagella/Pilli by EM??. Based on the experimental design there is no direct scientific evidence that the negative effects of the CFS on Pseudomonas sp. growth is due to flagella/Pilli destruction.
11. I noticed the usage of different types of media/different suppliers in the same/different experiments. The produced antimicrobial agents are expected to differ by doing so even for the same bacterial strain. For example, the produced volatile fatty acids will differ after growing the same bacterial strain on MRS 172 broth or HiVeg MRS broth, India. Moreover, it is very important to have "media only" as control in this experiment. I can see in Table 3 the control media results but there is no mention in methods about any controls.
12. Line 227>> Scientific names should be always in italic. Check ALL the manuscript.
13. Figure 1, the gel image is not clear; the red labels are not readable.
14. In legend of Figure 2>> were 'done' in triplicate.
15. In Figure 2>> Why some histogram bars have SD and others are not?
16. Please comment on Figure 2 A isolate L1 zone of inhibition?. According to the histogram, the neutralized CFS has more activity than the untreated CFS (acidified CFS)?? What could be the reason?
17. In Figure 2, have you tested the milk and soymilk media only effects? I mean the milk and soymilk may have some inhibitory effects even without bacterial growth.
18. In Figure 3, define "E" column?
19. In Figure 3, SD bars are missing for some histogram bars.
20. In Figure 7, the histogram shows that L2 has almost 18 mm zone of inhibition. According to the histogram, L2 is more effective than L1. I don't see this big zone (18 mm) on the plate (Figure 7, B) and L1 sounds to have even a bigger zone of inhibition.
Based on the previous comments, I don't think the manuscript is publishable in the current format.
Comments on the Quality of English LanguageThe English language of the manuscript needs revision.
Author Response

(The authors gave the same response as above.)

Reviewer 3 Report
Comments and Suggestions for Authors
The authors should better explain the purpose of the study and in what way or field the selected lactobacilli can be used as postbiotics or prebiotics
The authors should report the OD600 obtained at the end of growth
The authors should report the CFU/ml obtained at the end of growth, before to perform the antimicrobial activity
After centrifugation, is the supernatant tested directly for the evaluation of antimicrobial activity or is it first concentrated via membranes? in this case it is necessary to report the details of the process.
page 15 line 538, 541 and 546 put the names of the microorganisms in italics, please.
Please, check in the document that all the names of the microorganisms are written in italics.
many references are old, add some more recent references
page 1 line 41 “e.g., the dynamics of fermentation”, please add two references: Lactobacillus brevis CD2: Lactobacillus plantarum: Microfiltration experiments for the production of probiotic biomass to be used in food and nutraceutical preparations; Fermentation Strategies and Extracellular Metabolites Characterization.
Comments on the Quality of English Languageno comments
Author Response

(The authors gave the same response as above.)

Round 2
Reviewer 2 Report
Comments and Suggestions for Authors
All my comments were addressed/clarrified by the authors.
Comments on the Quality of English LanguageMinor editing/revision of English language is required.
Author Response
DEAR Editor,
Thank you very much for your time and for your evaluation of our manuscript microorganisms-2693066, entitled: From traditional dairy product "Katak" to useful strains of Lacti-plantibacillus plantarum.
Once again, we would like to thank the esteemed reviewers for their consideration and helpful recommendations. Following the assessment by reviewer 2, the English language and grammar in the written article were carefully edited. The invaluable assistance we received from Alexis Manaster Ramer, PhD (former Professor Wayne State University, USA ) and Diana Bogoeva, PhD, (Curtin University, Australia).
We sincerely hope that we have fulfilled all the requirements with the current version of our manuscript. We have prepared it as per the deadline (of 5 days) , mentioned in your kind letter.
Please find the edited manuscript uploaded.
Looking forward to hearing from you, we remain with best wishes.
Respectfully, on behalf of the author team, Svetla Danova (Corresponding Author)
